# The Economic Value of Water Ecology in Sponge City Construction Based on a Ternary Interactive System

**DOI:** 10.3390/ijerph192315844

**Published:** 2022-11-28

**Authors:** Wenzhao Zhou, Yufei Wang, Xi Wang, Peng Gao, Ciyun Lin

**Affiliations:** 1School of Management, Jilin University, Changchun 130025, China; 2Dalian Branch, China Development Bank, Dalian 116001, China; 3Economic Information Center of Jilin Province, Changchun 130061, China; 4Economic Research Institute of Jilin Academy of Social Sciences, Changchun 130032, China; 5Qingdao Transportation Public Service Center, Qingdao Municipal Transport Bureau, Qingdao 266061, China; 6School of Transportation, Jilin University, Changchun 130022, China

**Keywords:** ternary interactive system, sponge city, water ecology, economic value

## Abstract

Ecological water resources occupy a vital position in the national economy; without sufficient ecological water resources, the construction and economic development of sponge cities would be seriously restricted. Appropriately, the Chinese government proposed that sponge city planning should be carried out in accordance with the number of available ecological water resources. The government therefore put forward the method of conservation and intensification to solve the problem of water shortage. This paper highlights the interactions between ecological water resources, sponge cities, and economic development in northern China, starting with the interaction and mechanism of action that concerns ecological water resource utilization, sponge cities, and economic development. In the empirical test, the dynamic changes of the three indicators were analyzed empirically using the panel data vector autoregression method, and the dynamic relationship of each factor was measured using generalized moment estimation. It was found that ecological water resources are a key factor in promoting regional economic development, and the relationship between ecological water resources and sponge cities is both supportive and constraining; therefore, the constraints that ecological water resources place on sponge cities also indirectly affects economic development. To disconnect the use of water and ecological resources from economic development, it is necessary to note the following: the feedback effect of economic development and the resolution of the contradiction between sponge cities, water, and ecological resource use.

## 1. Introduction

With the rapid development of the economy and the shortening of the product life cycle, the rate of product renewal has quickened. This has been exacerbated by global energy shortages and increasing concerns surrounding environmental protection in various countries [1,2,3,4]. Water resources are important as they allow people to survive on Earth, and the water ecosystem not only provides the basis for human survival and development, but, to a certain extent, it also maintains the planet’s ecological structure, ecological environment, and ecological functions. In recent years, the Chinese economy has developed rapidly, and the process of sponge city planning has been accelerated as the phenomenon of ecological water resource wastage has become increasingly serious and ecological water resource shortages have greatly impacted people’s lives. Given the present situation regarding ecological water resource shortages and the deteriorating water environment, coordinating the rational development and utilization of ecological water resources, and enhancing the service function of the water ecosystem, have become urgent tasks [5,6,7,8,9]. The assessment of the service value of the water ecosystem can help people to better understand the value of ecological water resources and promote the sustainable development of ecological water resources [10,11].

Zhang et al. established a multi-level evaluation index system with Gui’an New Area as the research object and they gave related policy recommendations [12]. Huang proposed a set of indicators such as “water ecology”, “water environment”, “water safety”, “construction system”, “implementation”, and “display degree”, and they established an evaluation model of a “sponge city” based on an “entropy method” and “TOPSIS”. Moreover, they conducted their evaluation in the city of Hebi [13]. Lei drew on a series of landscape performance research results in the United States. Based on the “Evaluation criteria for sponge city construction”, Lei systematically sorted out and summarized the evaluation indexes of sponge cities in China [14]. After collating the theoretical studies and references on sponge city construction in China, Jia found that the theoretical studies in China account for all the available categories that can be studied; however, there are still deficiencies regarding character studies and effective technology development, which need to be supplemented and improved urgently.

Based on the above research results, this paper used factor analysis and fuzzy hierarchy analysis using “sponge city construction evaluation criteria”. This was combined with the actual situation in the provinces and cities in the north of China, which enabled the creation of a sponge city performance evaluation method [15,16,17,18,19]. Then, the dominant index that reflected the effects of sponge city construction was selected, and the fuzzy hierarchical analysis method was used to evaluate the changes to the water environment in the northern region between 2016 and 2020. The ecological water environment after the implementation of “sponge city” pilot was thus optimized [20,21,22,23].

## 2. Research Hypothesis

### 2.1. The Influence of Ecological Water Resources on Economic Growth

With the development of the social economy, the demand for ecological water resources is increasing, and in the process of sponge city construction, an increasing number of people start to concentrate in the city, thus increasing the amount of water that is used. Ecological water is the most important link in the urban ecosystem, as it can protect the urban ecological environment, reduce the urban heat island effect, and purify the air [24,25,26,27,28,29,30,31,32,33,34,35]. There is a close link between the use of ecological water resources and manufacturing processes; indeed, water plays an irreplaceable role in industrial production, as nothing can occur without using water. The amount of water used due to industrial production is high, second only to agricultural water use [36]. In 2017, the total industrial output value of the country reached 28 trillion yuan, and industrial water consumption was more than 127.7 billion cubic meters; this fully illustrates the huge supporting role that ecological water resources play with regard to industrial production. Due to the limited number of water ecological resources, the supply usually does not vary significantly, but as industries continue to develop, the demand for water quality and quantity will increase accordingly. The gradually increasing elasticity of the supply of ecological water resources will not be able to match the development rate of cities and economies, and the increase in the level of demand for ecological water resources will also have a negative impact on the economy of our country. Moreover, to some extent, it will also affect the supply of ecological water resources. At the same time, the difference in regional natural conditions has led to an imbalance in the development of ecological water resources and the economy; this has a higher degree of impact than the level of economic development between regions. Therefore, places with a weak economic base and abundant ecological water resources have minimal constraints on economic development, whereas places with a better economic base but a relative lack of ecological water resources will encounter more ecological water resource shortages. When the ecological water resources cannot meet the water demand of the region, it will hinder economic development to a certain extent. On this basis, the following assumption is given in this paper.

**Hypothesis 1** **(H1).**
*The use of water ecological resources has a positive impact on economic growth.*


### 2.2. The Impact of Sponge City Construction on Economic Growth

In recent years, with the continuous growth of the urban population, and the continuous promotion of sponge city construction, if the city has not been planned sufficiently enough, “spreading the pie”, the creation of an “urban village”, “resource depletion”, “environmental pollution”, and other problems, will make the management costs and the cost of social living rise significantly. If there is chaos when constructing a sponge city, it will cause huge economic losses, and stable and sustainable economic development will not be maintained [37,38]. Although sponge cities are the main driving force of economic development, the problems they face will become increasingly prominent as the level of urbanization increases; therefore, if urbanization affects cities more than sponge cities, this will also hurt the development of the economy. The current priority is to solve the problems concerning “sponge cities”, reduce urban congestion and resource shortage, improve urban life and ecology, narrow the gap between rich and poor, solve the conflicts in medical care and education, and achieve harmony between sponge cities and economic development [39]. In the process of economic development, all relevant departments should have more funds to promote the sponge city process. In the current process of sponge city construction, due to the lack of a single source of funding (which results in bottlenecks in sponge cities), many sponge city processes do not take into account people’s livelihoods, thus resulting in a series of urban problems, such as the creation of a “half sponge city” or a “passive sponge city”. Moreover, economic development can help avoid the shortage of funds that may occur during the sponge city construction process, which would provide financial assurance during this construction process. It can also provide long-term security to the lives and living conditions of vulnerable groups such as migrant workers, which highlights the inclusiveness and harmonious nature of sponge cities. On this basis, the following assumption is given in this paper [40,41,42,43].

**Hypothesis 2** **(H2).**
*Sponge cities and economic development are influenced by each other.*


### 2.3. The Impact of Sponge City Construction on the Use of Ecological Water Resources

Ecological water resources are central to urban development, and sponge cities mean a more dense urban population and an increased demand for ecological water resources in the same area; therefore, as the level of development of sponge cities in China continues to increase, water resources have come under increasing pressure. Astronomers believe that the world is similar to a large sponge; each planet revolves around this sponge, and the weight of the stars have the ability to suck a large hole out of the sponge. If the volume of a planet is infinite, then this area will become a bottomless abyss, and any object close to it will be sucked in, to the point where even light cannot escape. The regional demand for ecological water resources is the same as the astronomers’ description of a black hole, because the uneven distribution of the population leads to an uneven demand for ecological water resources; a rural area is similar to a planet, a city is similar to a star, and a huge metropolis is similar to a black hole. Although the ecological water resources can meet needs according to the average water consumption level of the population, in some places, such as cities, the pressure on the supply of ecological water resources is very high, thus leading to a continuous shortage of local ecological water resources. Sponge city construction treats ecological water resources as important natural resources, and a large proportion of the domestic and ecological water supply can be used jointly, thus improving the utilization of ecological water resources and maintaining or reducing regional water consumption; however, the pressure on ecological water resources within cities is growing, and high-intensity urban construction has changed the original ecosystem, thus leading to urban nonpoint source pollution, which causes rivers to deteriorate in terms of water quality and ecological function. As the country has increased its industrial production output, nonpoint source pollution has gradually risen to become the primary source of pollution with regard to ecological water resources; this is adding to the crisis concerning ecological water resources in cities. The increasing population and the growing industrial economy has led to a dramatic increase in the use of ecological water resources in local areas. Technological progress can improve the utilization rate and reduce the consumption of ecological water resources, but the shortage of ecological water resources in cities still exists because technological progress cannot obtain a great deal of progress in the short term; therefore, effectively alleviating the ecological water resource tension in local areas has become an urgent matter. In addition, the deterioration of water ecology and the water cycle in cities, as well as the expansion of cities, has led to the overuse of the water environment and groundwater. The expansion of cities has led to the expansion of non-permeable areas in cities, the renovation of sewers has led to changes in the drainage system, the expansion of the land area has led to the reduction of the water surface and green areas, as well as an increase in water evaporation, surface runoff, and water pollution. On this basis, the following assumption is given in this paper.

**Hypothesis 3** **(H3).**
*The development of ecological water resources is an important support for, and constraint on, the development of sponge cities.*


## 3. Research Design

### 3.1. Data Sources

In this study, the panel data of 15 provinces in northern China from 2016 to 2020 were selected as the research object, and the data mainly came from the China Environmental Statistical Yearbook and China Statistical Yearbook. In order to eliminate heteroscedasticity, the logarithms of absolute exponents in the model were processed. Water resource utilization (water consumption) became Inwater, the urbanization level became Inurban, and economic development became Ingdp.

In this empirical study, the relevant indicators are defined as follows. (1) The utilization of ecological water resources, according to academic tradition, is calculated by the total amount of water used every year; however, given that this paper focuses on sponge cities, the relationships between the utilization of ecological water resources, sponge cities, and economic development are studied instead. Nevertheless, it must be noted that agricultural water is one of the most important ecological resources because it has environmental benefits to urban areas. Although agricultural water consumption has not increased significantly, it is necessary to retain agricultural water as an ecological water resource. (2) The comprehensive assessment of sponge cities adopts the current mainstream sponge city index, that is, the proportion of urban population in the total population. (3) At the economic level, a proportion of agricultural water is reserved when selecting the index of ecological water resources; therefore, all industrial output value is reserved in the GDP index.

### 3.2. Research Methods

In order to further verify the interaction structure underpinning water resource utilization, a city’s urbanization level, and economic development, the panel data of 15 provinces, municipalities, and autonomous regions between 2016 and 2020 were used to verify the interactions between the three variables.

The characteristics of the panel vector autoregressive model are as follows: the model not only reduces the internal problem effectively, but it also effectively controls the differences found in the cross-section. Before PVAR regression, the unit root test must be applied to check the stability of the data to prevent pseudo-regression. In this paper, the LLC method, IPS method, ADF-Fisher method, PP-Fisher method, and four other panel data cell root test methods were adopted. The stability was determined in accordance with most cases. Based on the element root test, the co-integration relationship among variables was analyzed by the Johansen test. After the optimal lag value was determined, the PVAR model was used for moment estimation, and the direction and magnitude of the interaction between the vectors were calculated. We also calculated whether these results were consistent with the predicted results of the ternary interaction system. Secondly, the influence of each variable on other factors during different time points was studied using impulse response analysis. Finally, using the method of variance decomposition, we investigated the response distribution of a variable during a specific time period, and we investigated the effects of each variable on a different variable.

PVAR has two different characteristics: dynamic delay and panel. This method can effectively overcome multi-collinear and endogenous problems, and it can also effectively control the individual characteristics of samples to ensure the correctness and unbiased estimation. The mathematical formula for PVAR is as follows:(1)$$y_{i,t}=\alpha_{i}+\beta_{0}+\overset{k}{\underset{j=1}{\sum }}\lambda_{j}y_{i,t}+r_{t}+\varepsilon_{i,t}$$
where $y_{i,t}$ is the matrix of endogenous variables, $\beta_{0}$ denotes the intercept term, i de notes the geographical unit, and the time unit is denoted by t. $y_{i,t-j}$ represents the matrix of explanatory variables, which is composed of lagged terms of endogenous variables, $\lambda_{j}$ is the estimation matrix of the lagged *j*th order, $\alpha_{i}$ denotes the regional fixed effect, $r_{t}$ denotes the time effect, and $\varepsilon_{i,t}$ is the random error term.

### 3.3. Model Design

Through the comprehensive evaluation of the above three elements, the results show that the development of ecological water resources in China is closely related to sponge cities and economic development, and all three elements have corresponding response mechanisms, which thus form a ternary interaction system. This is illustrated in Figure 1, which is an interactive diagram showing the relationships between ecological water resource use, sponge city construction, and economic development.

By establishing a model whereby the interactions between ecological water resources, sponge cities, and economic development are shown, the role of ecological water resources in sponge cities and regional economic development, as well as the operation and direction of the whole system, can be more clearly understood [44,45,46]. Ecological water resources are important resources that are indispensable for economic development, and the supply of water resources is essential in the process of economic development. If regional ecological water resources cannot meet the needs of economic development, and when the carrying capacity of ecological water resources exceeds a certain limit, it causes the ecological water resources in the region to be damaged or to verge on depletion, meaning that it can no longer accommodate a larger population. At the same time, economic development also promotes the use of ecological water resources, with regard to industrial production and regular daily activities. Through the continuous accumulation of capital, and with advances in technological development, the effective utilization of water resources can be improved, thereby reducing the demand for water resources and the pressure on the water supply. This should reduce the lag effect on the water supply in order to achieve a positive interaction between the use of water resources and economic development [47,48,49].

## 4. Empirical Analysis

### 4.1. Unit Root Test of Panel Data

In this paper, six aspects of annual data concerning 15 provinces and cities in northern China between 2016 and 2020 are analyzed in detail. These six aspects are: total water resources, total water supply, total ecological water resources, per capita water resources, urban population ratio, and GDP index. These six aspects were used with the aim of providing data reference for the following empirical analysis. See Table 1 for details.

Since panel data have commonalities in terms of time series and profiling data, they comprise a collection of two types of information. When building a time series data model, a unit root test is performed to prevent pseudo-regression, and to ensure the stability of this model, a unit root check is performed on this model. In this study, unit root tests were conducted for each panel datum using the IPS test, LLC test, ADF Hsher test, and PP-Fisher test. In the unstable original series, one difference was performed on the original series, the segmentation and trend terms were checked, and the results showed that after one difference, excluding trend and distance, the majority principle could be used for the final smoothness decision. Here, the check procedure was completed by Eviews v7.0 statistical software (QMS (Quantitative Micro Software, LLC), Irvine, CA, USA), and its check results are as follows (see Table 2).

The test results showed that the Ingdp and Inevaluation index *p*-values were both 0 and below 0.05, thus negating the original assumptions. The Inwater index was above 0.05 in both the IPS and ADF-FISHER tests, both of which indicated that the original assumptions could not be disproven and that there should have been a unit root; thus, instability was present. The unit root test was performed again after the first-order difference treatment of the data, and it was found that all four tests did not admit a unit root, thus indicating that the first-order difference data were a first-order stable series; therefore, PVAR analysis was performed using DIn-water, DInevaluation, and Dingdp to prevent them from generating pseudo-regressions.

### 4.2. Panel Data Cointegration Test

In the panel unit root test, the raw data were differenced once, and the results were all smooth. The Johansen co-integration test among variables could have been conducted to verify the long-term stable equilibrium relationship between the level of change regarding water use in sponge cities and economic development (see Table 3).

The results of the Johansen cointegration test are shown in the Table 2. The *p*-value exceeds 0.05, thus indicating that the original assumptions have been adopted and the three variables DInwater, DInvaluation, and DIngdp have been cointegrated; therefore, the three variables show a stable equilibrium relationship over a longer period of time.

### 4.3. Determination of the Optimal Delay Time

The autoregressive model of the panel vectors was estimated using the PVAR2 package, and the optimal lag order of the variables was evaluated using the Akira Pool Information Criterion (AIC), Bayesian Information Criterion (BIC), and Hannan-I Quinn Information Criterion. Through the comprehensive evaluation of the AIC, BIC, and HQIC criteria, the PVAR model of the fifth order lag stage was finally selected. The results are shown in Table 4.

### 4.4. Panel Moment Estimation

With the PVAR2 model, the GMM is used for estimation by default. Before GMM estimation, the fixed and time effects must first be eliminated by using the “forward mean difference” (HeImert) method, then, the time effects are removed using the mean difference method. The final GMM estimates are shown in Table 5 after the estimation is completed.

First, the development rate of the use of ecological water resources were chosen as the relevant variables, where the lag one and lag three of the two variables were independent variables, and the lag three factors of water use were all positive. The delayed period had a significant effect on water use in the earlier period, with a result of 5% at a factor of 0.1009; this indicates that the growth of water use in the earlier period promoted the growth of water use in the later period, thus indicating that the water consumption of the region has a “ratchet effect”. Moreover, given the current situation, water use will not decline rapidly and will rise at an increasing rate [50,51,52]. The sponge city with three lags had a positive effect on the growth rate of water consumption; this was significant with a result of 1% and an impact factor of 0.0371, thus indicating that the increase in urbanization rate promoted an increase in water consumption. The long-term lag in sponge city development promoted the growth of water resources; this is a reflection of the surge effect of water consumption due to population concentration and industrial agglomeration during sponge city construction. In later stages, economic development had a significant negative impact on the growth of water consumption, with an impact factor of −0.2555 at the 10% level; the three lags had a positive effect on the increase in water consumption, with a significant impact at the 5% level, thus indicating that in the early stages, rapid economic development had a certain promotional effect on the increase in water consumption, and the coefficient in the later stages was negative. This indicates that the supply of water resources in the north of China is insufficient and cannot adapt to the needs of economic development; on the other hand, the feedback effect of economic development on water resources makes water resource use more efficient, which, in turn, reduces the rate of water consumption. The amount of water used therefore remains balanced or it decreases [53,54,55,56]. The second factor is the rate of sponge city development as the dependent variable, with the lagged one-period to lagged three-period values of the three variables as the independent variables; the growth of water volume in all three lagged stages was −0.0082, −0.004, thus indicating that the current growth rate of water demand in northern cities is too fast. This is an unfavorable outcome for sponge city development as the constraining effect of water resources in the sponge city process is greater than the supporting effect, thus hindering the development of sponge cities. The effect of speed on sponge city construction was negative and the role factor of lag phase II is −0.0449, which is significant at a 1% confidence level; this indicates that the speed of sponge city construction in the northern region will gradually slow down, and the construction of sponge cities will become increasingly difficult as the number of sponge cities increases. Finally, economic development was taken as an influencing factor, and the lagged one-period to lagged three-period values of each variable were used as independent variables. The contribution of water consumption to economic growth for lag one and lag two was positively correlated, with corresponding coefficients of 0.0666 and 0.0913, respectively; these are significant at the 5% level, thus proving that the effect of water resources in terms of reducing economic growth is better than the blocking effect of water resources on economic growth, and the support effect of lag two is better than that of lag one. The impact factor of sponge city speed on economic growth in lag three is also positive, with a coefficient of 0.2160; this is significant at a 5% confidence level, thus proving that sponge city construction also contributes significantly to economic growth, mainly because of the scale effect of urban industry and the modernization of agriculture in the countryside. Moreover, this contributes significantly to the economic development of the northern region [57,58,59,60,61,62,63].

After completing the preparatory work for the preliminary empirical analysis, the estimated coefficients of the panel moments were obtained through the analysis of the PVAR2 model. The empirical study found that the sponge city construction process correlates with water consumption growth, with a coefficient of 0.0371; this reflects the supporting and restricting effect of effluent ecological resources on economic development. The growth rate of water use in the lagged phase I and lagged phase III were both negative, thus indicating that the current growth rate of water use in northern cities is too fast; this has become an unfavorable occurrence during the development of sponge cities. Moreover, the ecological water resources in sponge cities are more inhibitory than supportive, thus hindering the development of sponge cities. The influence factors of economic development on urbanization development in lag phase II and lag phase III were 0.0317 and 0.0158, thus highlighting the reduced economic development in sponge cities. The contribution of ecological water resources to economic growth in both lag phase I and lag phase II have a significant positive relationship, thus indicating that ecological water resources play a more significant role in reducing economic growth. The rate of sponge city growth in the three lag phases has a positive effect on economic growth, thus proving that the boosting effect exists in terms of increasing the rate of sponge city growth, and that it has a positive impact on economic development. Based on the theoretical results from the literature, and combined with the results of the empirical analysis, this paper puts forward the following recommendations.

## 5. Conclusions and Recommendations

### 5.1. Conclusions

As a major type of ecosystem, water ecosystems not only provide the material basis and natural conditions for human survival and development, but they also maintain the structure, ecological processes, and ecological environment of the ecosystem. With the development of China’s economy and society, and given the acceleration of the urbanization process, the phenomenon of the over-exploitation and utilization of water resources has become increasingly prominent, and the impact of human activities on the water ecosystem has worsened. In the face of water resource shortages and water environment deterioration, it is crucial to coordinate the effective development and utilization of water resources and improve the service functions of the water ecosystem. This paper demonstrates the relationships between ecological water resources, sponge city construction, and economic development in northern China, starting with an examination of the role that the relationship between ecological water resource utilization, sponge city construction, and economic development plays. This paper empirically investigates the relationship between the three variables using panel data from 2011–2020 for fifteen provinces, cities, and autonomous regions in China. The results show that there is a close relationship between ecological water resource development, sponge city construction, and economic development in China which forms a triadic interactive system. In practice, it has been shown that water resources are effective in driving economic growth, and they do not tend to behave as deterrents to economic growth. The construction of sponge cities can promote economic growth, and the scale effect of cities and the modernization of agriculture and rural areas greatly promote the economic development of northern regions.

### 5.2. Recommendations

Insist on the technical innovation of the “sponge mechanism”. From the “low-pollution development” in the United States to the suggested “sponge mechanism”, the reform of integrated management and sustainable development of urban water resources has not stopped in the past few decades, although the ideas and technologies behind these reforms have changed tremendously. Economic development can promote the development of science and technology, and the progress of science and technology can also promote the use of water resources; therefore, it is necessary to strengthen scientific and technological innovations. In our opinion, we need to transform urban development and constructions into “sponge mechanism” technologies so that the urban water supply can keep up with sponge city development, urban water efficiency can be further improved, urban water demand can be stabilized, and water utilization and economic growth can finally be decoupled.Adhere to the concept of “integration”. Coordinate and promote the construction of a “sponge city” to meet the developmental needs of the modern period; develop a set of construction technologies and technical standards; and combine these factors in accordance with local conditions in order to achieve the individualization of sponge city construction. Comprehensive consideration of urban water conditions, the climate environment, rainwater and sewage cycles of the city, the urban economy, optimization of urban drainage control levels, and the combination of sponge city specific module facilities must be ensured. Artificial intervention and natural regulation can aid with the development of sponge city and improve the quality of the city.Construction of ecological sponge city should be close to nature. Based on the concept of ecological economy and sustainable development, the development of the city should inevitably integrate humans and nature; therefore, during the construction of sponge cities, it is necessary to follow nature and let nature play its role. Let lakes and rivers have their own water storage function, and let green areas and gardens play their proper function, both to achieve the purpose of environmental protection and energy retention, and to realize that water resources can be recycled.Adhere to the leading role of the government. As the main body leading sponge city construction, the government must be unified; governments at all levels should increase support for sponge cities and further improve the sponge city construction standards, technical specifications, and the full implementation of sponge city planning. PPP financing can be used to invest in sponge city construction, accelerate technological innovation in water environment management, and promote the coordinated development of sponge cities and regional economies in the northern region. Finally, supporting policies relating to taxation, credit, capital, talent training, and talent introduction should improve the technical innovation ability of some large wastewater enterprises, especially regarding the effectiveness of wastewater treatment equipment.In order to maintain the water ecological environment, we should start by assessing the overall situation, then comprehensively analyzing water quality, and finally, we should use advanced technologies and ideas to provide a basis for protecting water resources. GIS technology has been widely used in water quality analysis, and data can be obtained effectively by monitoring hydrological data at various stages. In general, taking reasonable ecological protection measures can achieve better ecological results. By increasing the area of vegetation, soil erosion can be effectively reduced, and groundwater precipitation can be increased, thus enabling an effective internal circulation of water resources. Climate change is an important factor leading to water ecological and environmental problems, and low carbon and environmental protection measures should be followed; the relationship between production and emission should be reasonably adjusted. Greenhouse gas emissions should be reduced to achieve harmony and unity between water bodies and the atmosphere.According to the ecological environment pollution status of water bodies, finding the source of pollution, analyzing the type of pollution, understanding the current situation of the region, and reasonably controlling the pollution problem is essential. Through a deep analysis of wastewater, we screened and controlled the wastewater that meets the national wastewater treatment requirements, reduced the discharge of wastewater, and improved the ecology of water resources. Drip irrigation technology has been widely used in agricultural irrigation to save water resources and improve the utilization rate of water resources, as well as to achieve the established goals of agricultural irrigation and ensure the safety of agricultural production. At the same time, it is necessary to reduce the use of pesticides and chemical fertilizers and to reduce harmful substances and trace elements in groundwater to improve water quality. The concerned authorities should not only control the production units but also coordinate the agricultural sector to clarify the relationship between the two. We should establish a conservation based on realistic needs, take corresponding countermeasures into account, raise people’s awareness of water ecology and the environment, and consciously practice the mindful usage of water resources in daily life to gradually conserve water resources. At the same time, we should reduce the negative impact of industrial activities on the ecology of water bodies and contribute to the conservation of water resources.

## Figures and Tables

**Figure 1 ijerph-19-15844-f001:**
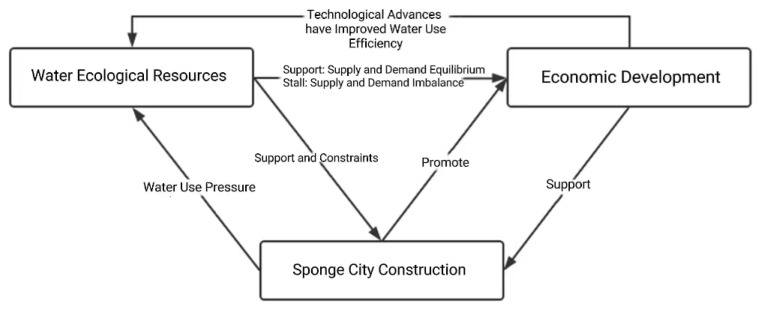
Schematic diagram of the ternary interaction system.

**Table 1 ijerph-19-15844-t001:** Data from 15 provinces and cities in Northern China between 2016 and 2020.

Provinces	Year	Total Water Resources (100 Million Cubic Meters)	Total Water Supply (100 Million Cubic Metres)	Total Ecological Water Use (100 Million Cubic Meters)	Aquatic Ecological Resources (Cubic Meters per Person)	Proportion of Urban Population (%)	GDP Index (100 Million Yuan)
Beijing	2016	35.1	38.8	11.1	161.6	86.74%	27,041.2
	2017	29.8	39.5	12.7	137.2	86.92%	29,883.0
	2018	35.5	39.3	13.4	164.2	87.09%	33,106.0
	2019	24.6	41.7	16.0	114.2	87.35%	35,445.1
	2020	25.8	40.6	17.2	117.8	87.5%	35,943.3
Gansu Province	2016	168.4	118.4	4.1	646.4	46.07%	6907.9
	2017	238.9	116.1	4.7	912.5	48.14%	7336.7
	2018	333.3	112.3	4.7	1266.6	49.70%	8104.1
	2019	325.9	110.0	5.2	1233.5	50.70%	8718.3
	2020	408.0	109.9	10.7	1628.7	52.2%	8979.7
Hebei Province	2016	208.3	182.6	6.7	279.7	53.87%	28,474.1
	2017	138.3	181.6	8.2	184.5	55.74%	30,640.8
	2018	164.1	182.4	14.5	217.7	57.33%	32,494.6
	2019	113.5	182.3	22.1	149.9	58.78%	34,978.6
	2020	146.3	182.8	29.9	196.2	60.1%	36,013.8
Heilongjiang Province	2016	843.7	352.6	2.5	2217.1	61.10%	11,895.0
	2017	742.5	353.1	1.5	1957.1	61.90%	12,313.0
	2018	1011.4	343.9	3.6	2675.1	63.45%	12,846.5
	2019	1511.4	310.4	1.2	4017.5	64.61%	13,544.4
	2020	1419.9	314.1	2.3	4419.2	65.6%	13,633.4
Henan Province	2016	337.3	227.6	13.0	354.8	48.78%	40,249.3
	2017	423.1	233.8	19.8	443.2	50.56%	44,824.9
	2018	339.8	234.6	23.6	354.6	52.24%	49,935.9
	2019	168.6	237.8	29.2	175.2	54.01%	53,717.8
	2020	408.6	237.1	35.0	411.9	55.4%	54,259.4
Ji Lin Province	2016	488.8	132.5	6.3	1782.0	58.75%	10,427.0
	2017	394.4	126.7	4.7	1447.3	59.70%	10,922.0
	2018	481.2	119.5	4.4	1775.3	60.87%	11,253.8
	2019	506.1	115.4	6.5	1876.2	61.64%	11,726.8
	2020	586.2	117.7	11.4	2418.8	62.6%	12,256.0
Liaoning Province	2016	331.6	135.4	5.6	757.1	68.87%	20,392.5
	2017	186.3	131.1	5.5	426.0	69.48%	21,693.0
	2018	235.4	130.3	5.7	539.4	70.26%	23,510.5
	2019	256.0	130.3	6.0	587.8	71.22%	24,855.3
	2020	397.1	129.3	7.4	930.8	72.1%	25,011.4
Inner Mongolia	2016	426.5	190.3	23.1	1695.5	63.38%	13,789.3
	2017	309.9	188.0	23.1	1227.5	64.61%	14,898.1
	2018	461.5	192.1	24.6	1823.0	65.52%	16,140.8
	2019	447.9	190.9	25.0	1765.5	66.46%	17,212.5
	2020	503.9	194.9	29.4	2091.7	67.5%	17,258.0
Ningxia Province	2016	9.6	64.9	2.0	143.0	58.71%	2781.4
	2017	10.8	66.1	2.5	159.2	60.99%	3200.3
	2018	14.7	66.2	2.6	214.6	62.11%	3510.2
	2019	12.6	69.9	2.8	182.2	63.60%	3748.5
	2020	11.0	70.2	3.7	153.0	64.9%	3956.3
Qinghai Province	2016	612.7	26.4	1.1	10,376.0	53.61%	2258.2
	2017	785.7	25.8	1.2	13,188.9	55.46%	2465.1
	2018	961.9	26.1	1.3	16,018.3	57.24%	2748.0
	2019	919.3	26.2	1.4	15,182.5	58.81%	2941.1
	2020	1011.9	24.3	1.1	17,107.4	60.0%	3009.8
Shaanxi Province	2016	271.5	90.8	3.1	713.9	56.40%	19,045.8
	2017	449.1	93.0	3.5	1174.5	58.07%	21,473.5
	2018	371.4	93.7	4.8	964.8	59.65%	23,941.9
	2019	495.3	92.6	4.5	1279.8	61.28%	25,793.2
	2020	419.6	90.6	5.2	1062.4	62.6%	26,014.1
Shandong Province	2016	220.3	214.0	7.6	222.6	59.13%	58,762.5
	2017	225.6	209.5	12.0	226.1	60.78%	63,012.1
	2018	343.3	212.7	10.6	342.4	61.45%	66,648.9
	2019	195.2	225.3	17.9	194.1	61.86%	70,540.5
	2020	375.3	22.5	19.1	370.3	63.0%	72,798.2
Shanxi Province	2016	134.1	75.5	3.3	365.1	57.26%	11,946.4
	2017	130.2	74.9	3.0	352.7	58.60%	14,484.3
	2018	121.9	74.3	3.5	328.6	59.85%	15,958.1
	2019	97.3	76.0	4.9	261.3	61.28%	16,961.6
	2020	115.2	72.8	4.8	329.8	62.5%	17,835.6
Tianjin Province	2016	18.9	27.2	4.1	121.6	83.30%	11,477.2
	2017	13.0	27.5	5.2	83.4	83.55%	12,450.6
	2018	17.6	28.4	5.6	112.9	83.95%	13,362.9
	2019	8.1	28.4	6.2	51.9	84.33%	14,055.5
	2020	13.3	27.8	6.4	96.0	84.7%	14,008.0
Xinjiang Province	2016	1093.4	565.4	6.5	4596.0	50.41%	9630.8
	2017	1018.6	552.3	10.2	4206.4	51.90%	11,159.9
	2018	858.8	548.8	30.5	3482.6	54.01%	12,809.4
	2019	870.1	587.7	49.0	3473.5	55.53%	13,597.1
	2020	801.0	570.4	46.2	3111.3	56.5%	13,800.7

**Table 2 ijerph-19-15844-t002:** Unit root test for panel data.

Inspection Method		Inwater	Inevaluation	Ingdp
Level Value	LLC	−3.50570	−6.06263	−16.1758
		−0.00204	(0.0000)	(0.0000)
	IPS	−0.31891	−5.57732	−8.26989
		−0.47424	(0.0000)	(0.0000)
	ADF-FISHER	44.91732	109.0238	120.3312
		−0.19788	(0.0000)	(0.0000)
	PP-FISHER	50.43972	106.0531	179.8452
		−0.08532	(0.0000)	(0.0000)
		DInwater	DInevaluation	DIngdp
First order differential values	LLC	−16.5546	−13.3675	−9.47654
		(0.0000)	(0.0000)	(0.0000)
	IPS	−9.66636	−7.56007	−4.81307
		(0.0000)	(0.0000)	(0.0000)
	ADF-FISHER	130.218	107.92116	85.30536
		(0.0000)	(0.0000)	(0.0000)
	PP-FISHER	156.2652	152.3628	140.4768
		(0.0000)	(0.0000)	(0.0000)

Note: All the above tests assume that Ho: has a meta-root, τ: test values are t-statistics, and the values in parentheses are *p*.

**Table 3 ijerph-19-15844-t003:** Table of panel data from the co-integration test.

Number of Covariance Equations	Fisher (Trace Statistics)	Prob	Fisher (Great Eigenvalue Statistic)	Prob
None	366.84	0.0000	333.84	0.0000
Atmost 1	95.004	0.0000	86.556	0.0000
Atmost 2	39.48	0.39204	39.48	0.3920

**Table 4 ijerph-19-15844-t004:** Determination of optimal lag period.

Number of Steps	AIC	BIC	HQIC
1	−9.87859	−8.72905	−9.41245
2	−9.68477	−8.26167	−9.10709
3	−12.9395 *	−11.2053 *	−12.2350 *
4	−12.7522	−10.6604	−11.9022
5	−12.2761	−9.76738	−11.2573

Note: * represents the optimal lag order under this information criterion.

**Table 5 ijerph-19-15844-t005:** Coefficients of panel moment estimation results.

Regression Coefficient	DInwater	DInevaluation	DIngdp
DInwater (L1)	0.1009 **	−0.0040 **	0.0666 **
DInevaluation (L1)	−0.0963	−0.0108	0.0336
DIngdp (L1)	−0.2555 *	−0.0145	0.3294 ***
DInwater (L2)	0.0686	−0.0004	0.0913 **
DInevaluation (L2)	−0.1970	−0.0449 ***	0.1036
DIngdp (L2)	−0.0151	0.0317 *	0.0562
DInwater (L3)	0.17124	−0.0082 **	0.0264
DInevaluation (L3)	0.0371 ***	−0.0260	0.2160 **
DIngdp (L3)	0.2647 **	0.0158 *	0.4305 ***

Note: *, **, *** indicate at significant levels of 10%, 5%, 1%, respectively.

## Data Availability

Not applicable.

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
