# Peer review of "The Economic Value of Water Ecology in Sponge City Construction Based on a Ternary Interactive System"

_ijerph, 2022, doi:10.3390/ijerph192315844_

Round 1
Author Response
Response to Reviewer 1 Comments
Point 1: The description of the paper is not clear. It only says that 15 provinces in North China are selected, but it doesn't specify which provinces? The article also does not say which data or indicators are used to reflect the construction of sponge cities?
Response 1: For the issue of not specifying which provinces you mentioned, I have added the corresponding data for 2016 to 2020 for the fifteen provinces and cities studied in the article and presented in Table 1 in the text. In this paper, we analyze the role of LID indicators in the construction of sponge cities.
Point 2: Not all cities in China have carried out sponge city construction, and it is not clear that weather the selected 15 provinces have carried out sponge city construction. Sponge cities are the main driving force of economic development What is the basis for this? What is the driving force of economic development for cities that have not built sponge cities? The matching between the research content and the research scope of this paper needs to be discussed.
Response 2: This paper uses factor analysis and fuzzy hierarchical analysis to propose a method for evaluating the performance of sponge cities, taking into account the actual situation in Beijing, to evaluate the changes in the water environment in the northern region from 2011 to 2020. According to the notice of the General Office of the Ministry of Finance, General Office of the Ministry of Housing and Urban-Rural Development, General Office of the Ministry of Water Resources, and other related documents, a total of 45 cities in China have carried out the construction of sponge cities. What is the basis for your statement that "sponge cities are the main driver of economic development? What is the driving force of economic development in cities that have not built sponge cities?" This question, first of all, the most direct part of the economic benefits generated by sponge cities is the benefit of using rainwater, and in some sponge city construction projects, the use of reuse operation of rainwater, sewage treatment, and drainage network facilities can generate a relatively stable flow of funds. The national treasury encourages sponge cities as an urban innovation strategy. Participating project companies usually generate revenue from three elements: project operation benefits, government purchases, and financial subsidies. In contrast, cities that have not undertaken sponge city construction, their economic development is usually driven by infrastructure development, development of science and technology, health and public utilities, and e-commerce platforms. For the issue of matching the content and scope of the study, the paper has been modified accordingly in the conclusion section.
Point 3: The research methods of this paper are mainly statistical and management methods to analyze panel data. The results and conclusions do not deeply analyze the relationship between the three, and do not show the expected mutual influence relationship.
Response 3: For this issue, we have made the corresponding changes in the conclusion.
Point 4: The structure of the paper is unreasonable, the conclusion and the discussion are together, how to distinguish the conclusion and the discussion content?
Response 4: As a result of the analysis, we have changed Part 5, “Conclusions and Discussions” to “Conclusions and Recommendations”.
Point 5: The introduction part needs to be improved, need to explain the status and progress of the research related to this paper, fully explain the necessity of this paper.
Response 5: Based on your suggestions, we have made corresponding changes to the introduction section to explain the current status and progress of the research in this paper, which fully illustrates the innovation and necessity of the paper. All modifications in this paper are marked in blue font.

Reviewer 2 Report
1. Line 168: Please show the proportion of different water sectors in the study area.
2. Line 176: Agricultural water use is excluded from this paper. However, agricultural water is one of important ecological resources due to its environmental benefits for urban area even though the agricultural water consumption does not increase significantly. It is necessary to consider the role of agricultural water as one part of water ecological resources.
3. Line 181. Could you show a study map of 15 provinces in northern China? It is better for readers to understand the site location in China.
4. Line 236: For empirical analysis, please show a table to list these three indicators: water ecological resources, proportion of urban population, and GDP index for 15 provinces.
5. Line 248: In Table 1, what kind of statistical software you applied in this study?
6. Line 334: The empirical study found that the sponge city process has a positive correlation with water use growth. However, sponge city is good idea for regulating or mitigating water use growth. This result is different. Why? Is it an urban development not a sponge city due to the indicator of urban population?
7. Line 371: Construction of ecological sponge city is close to nature. So it is necessary to follow nature and let nature play its function, especially for reducing the water use growth.
Author Response
Response to Reviewer 2 Comments
Point 1: Line 168: Please show the proportion of different water sectors in the study area.
Response 1: Thank you for your valuable suggestions. Compared with the proportion of the water department, the author thinks that the water resource consumption is more in line with the needs of this paper, so we add the data of three different kinds of water consumption.
Point 2: Line 176: Agricultural water use is excluded from this paper. However, agricultural water is one of the important ecological resources due to its environmental benefits for the urban area even though agricultural water consumption does not increase significantly. It is necessary to consider the role of agricultural water as one part of water ecological resources.
Response 2: Regarding line 176, you mentioned that "this paper does not include agricultural water use, and it is necessary to consider the role of agricultural water use as part of the water ecological resources", we mentioned earlier that in this paper, we have studied the relationship between water ecological resources use, sponge cities and economic development from the perspective of sponge cities. However, agricultural water use did not increase significantly , but declined, and the growth of non-agricultural water use contributed more to the overall water use of PU, which is why agricultural water use was excluded from this paper.
Point 3: Line 181. Could you show a study map of 15 provinces in northern China? It is better for readers to understand the site location in China. Line 236: For empirical analysis, please show a table to list these three indicators: water ecological resources, proportion of urban population, and GDP index for 15 provinces.
Response 3: Thank you for your valuable suggestions. The author has clearly shown the specific names of the 15 provinces in Table 1 to show relevant data. In response to your question about adding maps, after consideration and trial, we decided to present it in the form of a table. Because by comparison, we think that the table format is more intuitive and can directly reflect the comparison of data between cities and individual cities. Moreover, for electronic maps, currently there is no legal English version of the domestic application.
Point 4: Line 248: In Table 1, what kind of statistical software you applied in this study?
Response 4: Thank you for your valuable suggestions. In this paper, the author uses blue font to mark the software used.
Point 5: Line 334: The empirical study found that the sponge city process has a positive correlation with water use growth. However, sponge city is a good idea for regulating or mitigating water use growth. This result is different. Why? Is it an urban development, not a sponge city due to the indicator of urban population?
Response 5: Because of the impact of the indicators of the urban population, this problem, because of the development of sponge cities, so it will also bring a corresponding increase in water consumption, but because of the different regional natural conditions, resulting in the imbalance of water ecological resources and economic development, so those places with a weak economic base rich in water resources, the constraints on economic development are very small.
Point 6: Line 371: Construction of ecological sponge city is close to nature. So it is necessary to follow nature and let nature play its function, especially in reducing the water use growth.
Response 6: For line 371, the construction of an ecological sponge city is close to nature, so we should follow nature and let nature play its function. We adopted your suggestion later and added the phrase "especially to reduce the growth of water consumption".

Round 2
Reviewer 1 Report
The first revision responded to each comment, however, some part of the revision did not achieve the expected effect. It is suggested to revise the paper continuously and invite other experts to review it again.
1. Authors added the corresponding data for 2016 to 2020 for the fifteen provinces and cities studied in the article and presented in Table 1. No direct relationship can be seen between Table 1 and the topic (LID or sponge city) of the paper. It is hoped that the author can provide data related to the research topic.
2. Comment said “Not all cities in China have carried out sponge city construction, and it is not clear that weather the selected 15 provinces have carried out sponge city construction.”
The author does not give a clear answer about the construction of sponge cities in the selected 15 provinces.
The author wrote that: ”taking into account the actual situation in Beijing, to evaluate the changes in the water environment in the northern region from 2011 to 2020.” Is the author going to use Beijing to represent the 15 northern provinces?
3. Comment “Authors said that: “sponge cities are the main driving force of economic development”. What is the basis for this? What is the driving force of economic development for cities that have not built sponge cities?”
Authors respond that “This question, first of all, the most direct part of the economic benefits generated by sponge cities is the benefit of using rainwater, and in some sponge city construction projects, the use of reuse operation of rainwater, sewage treatment, and drainage network facilities can generate a relatively stable flow of funds.”
The utilization rate of rainwater in Chinese cities is very low at present. It is not reasonable to say that the main economic benefits of cities depend on the utilization of rainwater. It is suggested that the author check it.
4. Last comment is “The structure of the paper is unreasonable, the conclusion and the discussion are together, how to distinguish the conclusion and the discussion content?”
Authors respond that “As a result of the analysis, we have changed Part 5, “Conclusions and Discussions” to “Conclusions and Recommendations””
The author did not understand the meaning of the comment. It is not just changed the title, but the content and structure need to be adjusted.
Author Response
Response to Reviewer
Comments
Point 1: Authors added the corresponding data for 2016 to 2020 for the fifteen provinces and cities studied in the article and presented in Table 1. No direct relationship can be seen between Table 1 and the topic (LID or sponge city) of the paper. It is hoped that the author can provide data related to the research topic.
Response 1: Based on your suggestions, we have modified Table 1 to make it more consistent with the research content of our topic.
Point 2: Comment said, “Not all cities in China have carried out sponge city construction, and it is not clear that whether the selected 15 provinces have carried out sponge city construction.”
The author does not give a clear answer about the construction of sponge cities in the selected 15 provinces. The author wrote that: ”taking into account the actual situation in Beijing, to evaluate the changes in the water environment in the northern region from 2011 to 2020.” Is the author going to use Beijing to represent the 15 northern provinces?
Response 2: According to the "General Office of the Ministry of Finance, General Office of the Ministry of Housing, Urban and Rural Development, General Office of the Ministry of Water Resources, Notice on the Demonstration Work of Systematic and Territorial Promotion of Sponge City Construction", the Notice on the Announcement of the Evaluation Results of the Provincial Work of Systematic and Territorial Promotion of Sponge City Construction Demonstration in 2021, and the "General Office of the Ministry of Finance, General Office of the Ministry of Housing, Urban and Rural Development, General Office of the Ministry of Water Resources, Notice on the Development of " The second batch of the "14th Five-Year Plan" systematization of the whole area to promote the demonstration of sponge city construction notice, China's public sponge city selected 45 cities: the first batch of 20 sponge city, A grade city 5: Shanxi Changzhi, Jiangsu Wuxi, Hunan Yueyang, Guangdong Province, Guangzhou City, Sichuan Province; B grade city 15: Tangshan City, Hebei Province, Jilin Province, Siping City, Jiangsu Province Suqian City, Zhejiang Province, Hangzhou City, Anhui Province, Maanshan City, Fujian Province, Longyan City, Nanping City, Jiangxi Province, Yingtan City, Weifang City, Shandong Province, Xinyang City, Henan Province, Xiaogan City, Hubei Province, Shantou City, Guangdong Province, Tongchuan City, Shaanxi Province, Tianshui City, Gansu Province, Urumqi City, Xinjiang Province. The second batch has 25 sponge cities, including Qinhuangdao, Jincheng, Hohhot, Shenyang, Songyuan, Daqing, Kunshan, Jinhua, Wuhu, Zhangzhou, Nanchang, Yantai, Kaifeng, Yichang, Zhuzhou, Zhongshan, Guilin, Guangyuan, Guang'an, Anshun, Kunming, Weinan, Pingliang, Golmud, Yinchuan. And the fifteen provinces in the northern region studied in this paper include Beijing, Gansu Province, Hebei Province, Heilongjiang Province, Henan Province, Jilin Province, Liaoning Province, Inner Mongolia Autonomous Region, Ningxia Hui Autonomous Region, Shaanxi Province, Shanxi Province, Shandong Province, Tianjin City, and Xinjiang Autonomous Region. All studied provinces have carried out the construction of sponge cities according to the relevant documents of sponge city construction in their provinces.
Regarding your question about whether the authors wanted to use Beijing as a representative of the fifteen provinces in the northern region for the study, we have deleted the content related to "taking into account the actual situation in Beijing, etc." in the last revision. The authors conducted the study by studying the data of 15 northern provinces, not by using the data of Beijing.
Point 3: Comment “Authors said that: “sponge cities are the main driving force of economic development”. What is the basis for this? What is the driving force of economic development for cities that have not built sponge cities?” Authors respond that “This question, first of all, the most direct part of the economic benefits generated by sponge cities is the benefit of using rainwater, and in some sponge city construction projects, the use of reuse operation of rainwater, sewage treatment, and drainage network facilities can generate a relatively stable flow of funds.”
The utilization rate of rainwater in Chinese cities is very low at present. It is not reasonable to say that the main economic benefits of cities depend on the utilization of rainwater. It is suggested that the author check it.
Response 3: Thank you very much for your suggestion. After careful consultation and consideration by our authors, we have decided to delete the phrase "sponge cities are the main driving force of economic development" in the abstract, probably because the English language or the expression is wrong and has caused you trouble. The original meaning is that sponge cities are one of the main driving forces of economic development. We are very sorry for the misunderstanding caused to you.
Point 4: The last comment is “The structure of the paper is unreasonable, the conclusion and the discussion are together, how to distinguish the conclusion and the discussion content?”
Authors respond that “As a result of the analysis, we have changed Part 5, “Conclusions and Discussions” to “Conclusions and Recommendations”
The author did not understand the meaning of the comment. It is not just changing the title, but the content and structure need to be adjusted.
Response 4: Thank you very much for your suggestion. In this revision, we have reorganized the structure and added the conclusion part of the article, and all the reworked parts are highlighted in yellow.
